# Patient-Reported Outcomes among people living with Chronic Pruritus (PROs-CP): Protocol for a single-center, multistage, mixed-methods prospective cohort study in Thailand

Surapon Nochaiwong[1,2]☯*, Chidchanok Ruengorn[1,2], Salin Kiratikanon[3], Rujira Rujiwetpongstorn[3], Panjit Chieosilapatham[4], Napatra Tovanabutra[3], Siri Chiewchanvit[2,3], Ratanaporn Awiphan[1,2], Chabaphai Phosuya[1,2], Yongyuth Ruanta[1,2], Kednapa Thavorn[1,2,5,6,7], Mati Chuamanochan[2,3]☯*

1 Department of Pharmaceutical Care, Faculty of Pharmacy, Chiang Mai University, Chiang Mai, Thailand, 2 Pharmacoepidemiology and Statistics Research Center (PESRC), Faculty of Pharmacy, Chiang Mai University, Chiang Mai, Thailand, 3 Division of Dermatology, Department of Internal Medicine, Faculty of Medicine, Chiang Mai University, Chiang Mai, Thailand, 4 Department of Microbiology, Faculty of Medicine, Chiang Mai University, Chiang Mai, Thailand, 5 Ottawa Hospital Research Institute, Ottawa Hospital, Ottawa, Ontario, Canada, 6 Clinical and Evaluative Sciences, ICES uOttawa, Ottawa, Ontario, Canada, 7 School of Epidemiology and Public Health, Faculty of Medicine, University of Ottawa, Ottawa, Ontario, Canada

☯ These authors contributed equally to this work.
* surapon.nochaiwong@gmail.com (SN); mati.c@cmu.ac.th (MC)

**Data Availability Statement:** No datasets were generated or analyzed during the current study, so

## Abstract

### Background

Although there have been well-validated patient-reported outcome (PRO) measurements in dermatology practice, there is limited evidence of the adopted comprehensive aspects of PRO measures in long-term follow-up among people living with chronic pruritus. As such, we aim to create a cohort study of the Patient-Reported Outcomes among people living with Chronic Pruritus (PROs-CP) in Thailand.

### Methods and design

This study is a single-center, prospective, open cohort, observational longitudinal study using a multistage, mixed-methods parallel designs to integrate both quantitative and qualitative data regarding PROs among people living with chronic pruritus (itch lasting six or more weeks). The multistage of the PROs-CP study will comprise three sub-studies: (i) study I, PROs measure development, translation, and psychometric validation; (ii) study II, perspectives of people living with chronic pruritus to gain more information regarding disease burden and unmet treatment care responses; and (iii) study III, a longitudinal study to assess the impact of chronic pruritus on long-term health outcomes. Based on a comprehensive review of a panel of stakeholders with chronic skin disease, a set of PRO measurement tools will comprise an established validated Thai version. Meanwhile, meaningful non-

the data availability policy is not applicable. However, all relevant data from this study will be made available under open access upon study completion.

**Funding:** This work was supported by Pharmacoepidemiology and Statistics Research Center (PESRC) and partially supported by Faculty of Medicine Research Fund (grant No. MED 74/2567), Chiang Mai University, Thailand. The funders had no role in study design, data collection and analysis, decision to publish, or preparation of the manuscript.

**Competing interests:** The authors have declared that no competing interests exist.

Thai versions or unestablished PRO instruments will be translated and developed through this study as appropriate. Quantitative data will be collected based on PRO measures regarding pruritus symptoms and severity, disease activity control and treatment satisfaction, general- and dermatology-specific health-related quality of life, mental health and psychosocial issues, and psychosomatic symptoms. Qualitative data will be obtained from the patient's perspectives through individual interviews.

## Ethics and dissemination

The study protocol was approved by the Ethics Committee of the Faculty of Medicine, Chiang Mai University (MED-2566-0299), Thailand. Our findings will be disseminated through scientific conferences and publications in peer-reviewed journals.

## Conclusion

Regarding the mixed-methods approach, this open cohort, prospective longitudinal study will provide an evidence-based better understanding of patient perspectives on chronic pruritus burden and inform the utility of a comprehensive set of PROs to measure their long-term health outcomes.

## Trial registration

Thai Clinical Trials Registry (TCTR, thaiclinicaltrials.org) registration TCTR20240327001 (registered on March 27, 2024).

## Introduction

Chronic pruritus is a common symptom in dermatology, characterized by the International Forum for the Study of Itch (IFSI) as present for more than six weeks [1, 2]. The point prevalence of chronic pruritus has been estimated to be more than one-third of patients in dermatology practice [3], and the lifetime prevalence rate ranges from 19.5% to 25.5% in the general population [4–6]. Chronic pruritus has substantial negative consequences on everyday life and deterioration of patients' health-related quality of life (HRQoL) [7–10]. Specifically, chronic pruritus is the major factor in relation to impairing the HRQoL and health outcomes among patients with chronic inflammatory skin diseases, systemic diseases, and idiopathic/rare diseases [7, 8, 10–13]. Furthermore, chronic pruritus has not only immense negative strains on HRQoL but also significantly impairs physical function and mental health issues, including healthcare utilization [8, 13, 14].

Unfortunately, there is no international consensus for objectively measuring tools to capture the severity and associated symptoms burden of pruritus in practice. To date, subjective symptoms experiences measured based on patient-reported remains the best available practical approach for quantifying disease burden, which accounts for and reflects the patient's view. However, apart from the clinical outcomes assessment, overarching aspects of chronic pruritus on a patient's life are underrecognized and poorly understood. Indeed, comprehensive sets of validated patient-reported outcome (PRO) measures are needed in dermatological practice to help identify and track responses to disease management treatments, clinically meaningful improvements, and support as a part of treatment decision-making. To close the gap and harmonize pruritus outcome measures, adopting standardized specific PROs that directly capture

perspectives of patients living with chronic pruritus in daily practice will facilitate better treatment comparison and outcomes reporting.

To the best of our knowledge, although there have been well-validated PRO measurement tools in dermatology research [10, 15–23], there is limited evidence of the adopted comprehensive aspects of PRO measures in long-term follow-up among people living with chronic pruritus. In this regard, we aim to create the PROs-CP (Patient-Reported Outcomes among people living with Chronic Pruritus), a single-center, open cohort, observational longitudinal study in Thailand. We will employ a multistage, mixed-methods approach to incorporate both qualitative and quantitative information from patient perspectives to better understand pruritus burden and associated symptoms and track treatment responses among Thai patients with chronic pruritus.

## Objectives

### Primary objectives

i.  To establish and validate the psychometric properties regarding validity, reliability, and responsiveness of the comprehensive sets of PROs-CP measures for Thai patients with chronic pruritus.

ii.  To explore the patient perspectives on living with chronic pruritus and the burden of pruritus severity and symptoms.

iii.  To assess the impact of chronic pruritus on long-term PROs-CP, including HRQoL, symptoms burden, mental health, psychosocial issues, and psychosomatic symptoms.

### Secondary objectives

i.  To define a minimal clinically important difference (MCID) of the particular PROs-CP measure for clinically meaningful interpretation among Thai patients with chronic pruritus.

ii.  To investigate the health utility and economic burden of Thai patients with chronic pruritus, including healthcare expenditures and out-of-pocket costs.

## Material and methods

### Patient and public involvement

Chronic pruritus patients will participate in the pilot testing and refinement of the questionnaire and provide feedback on the long-term acceptability of using the PROs-CP tools in daily practice. However, no patients or members of the public had a role in the study design, conceptualizing of the study, conducting the research, or reporting and dissemination plans.

### Study setting and design

This study is a single-center, prospective, observational longitudinal study at dermatology/general medicine clinics (public), Maharaj Nakorn Chiang Mai Hospital, Faculty of Medicine, Chiang Mai University, Thailand. The PROs-CP cohort will be conducted using mixed-methods parallel designs, integrating quantitative and qualitative data [24]. Based on a multi-methodical (Fig 1), the multistage of the PROs-CP study will comprise three sub-studies,

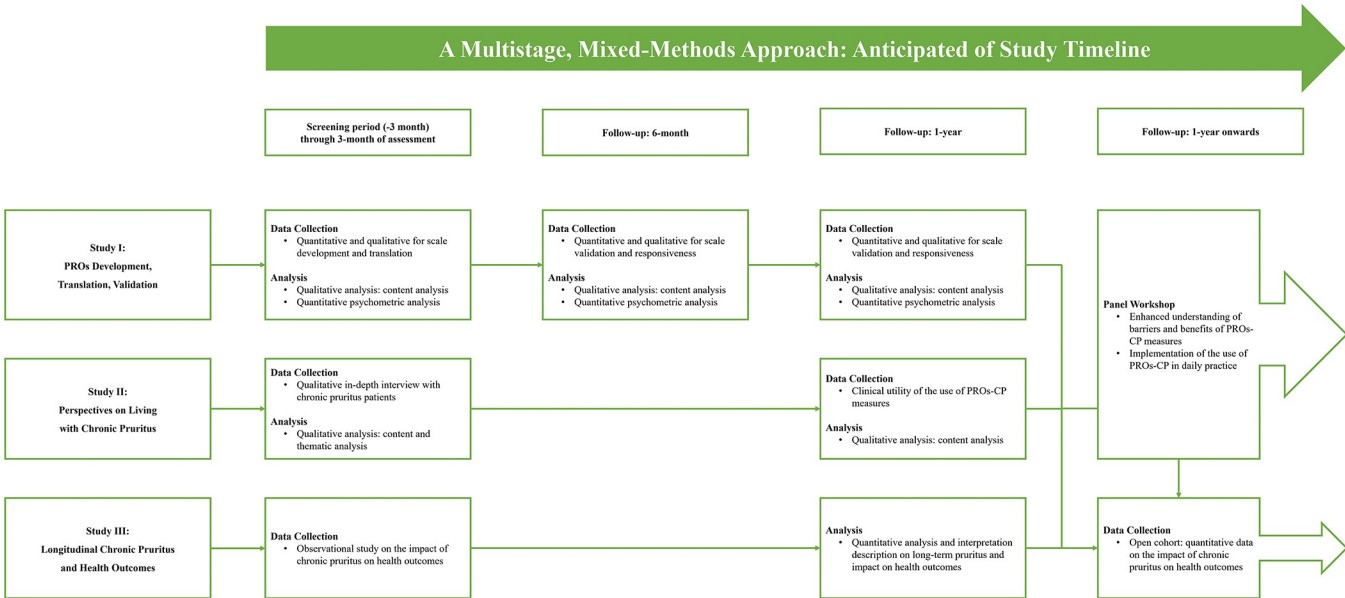

**Fig 1. Component of sub-studies and process of the PROs-CP cohort.** Abbreviations: PROs, patient-reported outcomes; PROs-CP, Patients-Reported Outcomes among People Living with Chronic Pruritus.

including study I, PROs measure development, translation, and psychometric validation; study II, explore the perspective of people living with chronic pruritus to gain more information regarding disease burden and unmet treatment care responses; and study III, a longitudinal study to assess the impact of chronic pruritus on long-term health outcomes (i.e., PROs, clinical outcomes and treatment responses, and health utilization and economic burden). Regarding the mixed-methods approach, this complex study process will provide an evidence-based better understanding of patient perspectives on chronic pruritus burden and inform the utility of a comprehensive set of PROs to measure their long-term health outcomes. This study was registered with the Thai Clinical Trials Registry (thaiclinicaltrials.org; TCTR20240327001, March 27, 2024).

## Study population

Based on routine clinical practice and to create an open or dynamic cohort, all adult patients with chronic pruritus, regardless of the IFSI classification to skin manifestation (i.e., inflamed skin, non-inflamed skin, and chronic secondary scratch lesions) in outpatient settings, will be consecutively recruited for this study project. Chronic pruritus patients will be diagnosed based on the recommendation of the IFSI—defined as pruritus lasting six or more weeks [1, 2]. Details of the eligible patients for the study are provided in Table 1.

## Study PROs measure instrument

The preliminary sets of PRO measures for patients with chronic pruritus were carefully reviewed based on comprehensive literature reviews and a panel of stakeholders with chronic skin, including dermatologists and experts in dermatology research (SN, CR, SK, RR, PC, NT, SC, and MC), health policy expert (KT) and patient's perspectives. Moreover, experts in qualitative research (RA, CP, and YR) in conjunction with a third-party expert in linguistics were also reviewed in terms of relevancy, readability, and applicability in the PROs measure to be included in the study. In this circumstance, most of the instruments in the aspects of pruritus

**Table 1. Eligibility criteria of the PROs-CP study.**

| Inclusion Criteria | Exclusion Criteria |
| --- | --- |
| • Patients aged ≥20 years at the date of screening<br>• Diagnosed with chronic pruritus based on the IFSI recommendation, itch lasting more than 6 weeks [1, 2]<br>• Could read and communicate in the Thai language<br>• Had an ability to understand and willingness to sign an informed consent statement and adhere to the study protocol | • Inability to answer questions or comply with the study protocol due to both physical and mental disability<br>• Recent hospitalization within 3 months<br>• The prognosis for survival <12 months |

Abbreviations: IFSI, International Forum for the Study of Itch; PROs-CP, Patients-Reported Outcomes among People Living with Chronic Pruritus.

symptoms and severity, general- and dermatology-specific HRQoL, mental health, psychosocial, and psychosomatic symptoms are validated and available in the Thai version, which is ready to implement in the study III. However, three validated non-Thai version instruments, namely—ItchyQoL (pruritus-specific HRQoL) [18, 25], Patient Unique Stigmatization Holistic tool in Dermatology (PUSH-D) [16], and Toronto Alexithymia Scale 20-item (TAS-20) [26, 27], which require to translate and testing for psychometric properties in this study. Moreover, we also identified unavailable and unestablished particular instruments for chronic pruritus regarding the domains of disease activity control and treatment satisfaction. Thus, we will develop and validate these instruments by conducting Study I. The PRO-CPs instruments ready to be used and planned to be translated and developed are described in Table 2.

## Study I: PROs measure development, translation, and psychometric validation

We will follow the established methods and consecutive phases for scale development to create multidimensional scales [8, 28]. The series of phases for scale development are illustrated in Fig 2. In brief, phase I—item generation will review the literature regarding general- and pruritus-specific disease activity control and treatment satisfaction. The subsample of 30 patients with chronic pruritus will be interviewed using structured and non-structured in-depth interviews to provide and reflect the perspectives regarding pruritus-specific disease control and their satisfaction with treatment care. In phase II, we will generate pilot questionnaires based on literature review and patient perspectives. The pilot questionnaires will then be reviewed by a panel of experts and patients to modify and verify in terms of face and content validity. In phase III, we will refine the particular questionnaire via an optimal sample to establish scale dimensionality using exploratory factor analysis (EFA, based on the rule of thumbs, 5–15 patients per question) and non-parametric item response theory (IRT) [8, 28, 29]. Lastly, in phase IV, we will perform an initial psychometric analysis to verify the validity and reliability of the final scale. For scale stability, a random subsample will be asked to complete the particular scale twice, within a 60–120 min interval.

Simultaneously, validated non-Thai version scales, namely—ItchyQoL, PUSH-D, and TAS-20 will be translated into the Thai-speaking language. The consecutive processes for scale translation are illustrated in Fig 1. Concisely, based on the permission from the authors of the original scales English version, the translation and cross-cultural adaptation will be performed in line with the standard guideline [30]. Two independent professional English-language translators will translate the measurement scale instructions and items from English into Thai. Of these, one person will be a board-certified dermatologist (MC), who is the study investigator and recognizes the study's purpose and objectives, while the other layperson will not

**Table 2. Measurement tools for the PROs-CP study.**

| Instrument | Established Measures | Available in the Thai Version | Description |
|---|---|---|---|
| **Aspect of Pruritus** | | | |
| Pruritus intensity—VAS | Yes | Yes | • A VAS-10 cm was used to measure the degree of pruritus intensity [12, 19].<br>• The recall period was determined to be 24 h, both the worst and average pruritus intensity. |
| Pruritus intensity—VRS | Yes | Yes | • A VRS based on a 5-point Likert scale was used to measure the degree of pruritus intensity [12, 19].<br>• The recall period was determined to be 24 h, both the worst and average pruritus intensity. |
| Pruritus intensity—NRS | Yes | Yes | • An NRS of 0 to 10 points was used to measure the degree of pruritus intensity [12, 19].<br>• The recall period was determined to be 24 h, both the worst and average pruritus intensity. |
| 5-D itch | Yes | Yes | • The 5-D itch scale consists of 8 items in 5 domains and evaluates itch symptoms.<br>• This scale revealed acceptable psychometric properties, with a Cronbach's α of 0.75 among pruritic dermatoses in Thailand [17]. |
| Patient's global assessment of pruritus control | No: developed through this study | Yes: psychometric testing through this study | • A single-item NRS of 0–10 points was used to measure the degree of pruritus activity control. |
| Global rating of change in pruritus symptom | No: developed through this study | Yes: psychometric testing through this study | • A single-item NRS of -7 (a very great deal worse) to 7 (a very great deal better) points was used to measure the degree of change in pruritus symptom over the past 4-week. If participants indicated that there had been no change, they were given a score of zero. |
| Chronic pruritus—multidimensional scale for disease control | No: developed through this study | Yes: psychometric testing through this study | • Planned to develop and initial validate a multidimensional scale for assessing disease activity control among patients with chronic pruritus. |
| Satisfaction with treatment care | No: developed through this study | Yes: psychometric testing through this study | • A single-item NRS of 0–10 points was used to measure global satisfaction with treatment care. |
| Treatment satisfaction-multidimensional scale for chronic pruritus | No: developed through this study | Yes: psychometric testing through this study | • Planned to develop and initial validate a multidimensional scale for capturing treatment satisfaction among patients with chronic pruritus. |
| **Aspect of HRQoL** | | | |
| DLQI | Yes | Yes | • The DLQI comprises a 10-item dermatology-specific HRQoL instrument (total score ranges from 0 to 30) [23].<br>• Internationally, this scale had acceptable to excellent psychometric properties, with Cronbach's α ranges of 0.75 to 0.92 [22].<br>• For banding of the score, DLQI was categorized into 0–1 (no effect), 2–5 (small effect), 6–10 (moderate effect), 11–20 (very large effect), and 21–30 (extremely large effect) [22]. |
| EQ-5D-5L | Yes | Yes | • The EQ-5D-5L measures health state utility values, 5-level assessment of mobility, self-care, usual activities, pain/discomfort, anxiety/depression, and the VAS of health state [44]. |
| Self-rated health—WHO | Yes | Yes | • A single item is used to measure QOL based on a 5-point Likert scale of self-rated health [45]. |
| ItchyQoL | Yes | No: translated and psychometric testing through this study | • The validity and reliability have been established on a satisfactory scale in individuals with chronic pruritus [18, 21, 25].<br>• The internal consistency and test-retest reliability (stability) with Cronbach's α reliability and intraclass correlation coefficient ranges from 0.89 to 0.99 and 0.84 to 0.98, respectively [18, 25].<br>• For scale interpretation, ItchyQoL scores were classified as follows: 0–30 (little), 31–50 (mild), 51–80 (moderate), and 81–110 (severe) [20]. |
| Global rating of change in HRQoL scale | Yes | Yes | • A single-item NRS of -7 (a very great deal worse) to 7 (a very great deal better) points was used to measure the degree of change in HRQoL. If participants indicated that there had been no change, they were given a score of zero [46]. |

(*Continued*)

**Table 2.** (Continued)

| Instrument | Established Measures | Available in the Thai Version | Description |
|---|---|---|---|
| **Aspect of mental health, psychosocial, and psychosomatic symptoms** | | | |
| PHQ-9 | Yes | Yes | • The PHQ-9 consists of 9 items to measure depressive symptoms (total score ranges from 0 to 27).<br>• This scale showed excellent psychometric properties, with a Cronbach's α of 0.90 among the general population in Thailand [29, 47].<br>• A cut-off point ≥9 was considered to indicate depressive symptoms [48]. |
| GAD-7 | Yes | Yes | • The GAD-7 consists of 7 items to measure anxiety symptoms (total score ranges from 0 to 21).<br>• This scale had excellent psychometric properties, with a Cronbach's α of 0.90 among the general population in Thailand [29, 47].<br>• A cut-off point ≥8 was considered to indicate depressive symptoms [49]. |
| WHO-5 well-being index | Yes | Yes | • The WHO-5 well-being index consists of 5 items to measure health-related personal well-being.<br>• This scale had excellent psychometric properties, with a Cronbach's α of 0.91 among the general population in Thailand [29, 47].<br>• A cut-off point <50 was considered a low well-being index [47]. |
| BRCS | Yes | Yes | • The BRCS consists of 4 items to measure cope tendencies with psychological stress (total score ranges from 4 to 20).<br>• This scale had excellent psychometric properties, with a Cronbach's α of 0.84 among the general population in Thailand [29]. |
| TAS-20 | Yes | No: translated and psychometric testing through this study | • The TAS-20 consists of 20 items to measure alexithymia (difficulty in identifying and describing emotions) [27].<br>• The non-Thai version showed adequate psychometric properties in 30 different languages (or dialects of the same language) [26]. |
| PUSH-D | Yes | No: translated and psychometric testing through this study | • The PUSH-D is a dermatologic-specific scale to evaluate stigmatization in patients with visible skin conditions, which consists of 17 items [16].<br>• This scale demonstrated excellent psychometric properties, with Cronbach's α 0.94 [16]. |
| RSES | Yes | Yes | • The RSES consists of 10 items to measure self-esteem and is widely used in social science research.<br>• The revised Thai RSES scale demonstrated excellent psychometric properties, with Cronbach's α ranging from 0.84 to 0.86 [50]. |
| UCLA Loneliness Scale-6 | Yes | Yes | • The short version of the UCLA loneliness scale consists of 6 items to gauge loneliness levels.<br>• This scale revealed acceptable psychometric properties in both non-clinical and clinical settings, with Cronbach's α ranging from 0.72 to 0.84 [51]. |
| ISI-7 | Yes | Yes | • The ISI-7 consists of 7 items to assess the nature, severity, and impact of insomnia (total score ranges from 0 to 28).<br>• This scale showed good psychometric properties, with a Cronbach's α of 0.84 among the general population in Thailand [29, 47].<br>• A cut-off point ≥12 was considered to indicate insomnia symptoms [52]. |
| SSS-8 | Yes | Yes | • The SSS-8 consists of 8 items to measure somatic symptom burden (total score ranges from 0 to 32) [53].<br>• This scale showed good psychometric properties, with a Cronbach's α of 0.84 among the general population in Thailand [29, 47].<br>• The SSS-8 severity categories were categorized as follows: 0–3 points (no to minimal), 4–7 points (low), 8–11 points (medium), 12–15 points (high), and 16–32 points (very high) [53]. |
| Global Sleep problems—NRS | No: developed through this study | Yes: psychometric testing through this study | • A single-item NRS of 0–10 points was used to measure the global degree of sleep problems. |

(*Continued*)

**Table 2.** (Continued)

| Instrument | Established Measures | Available in the Thai Version | Description |
|---|---|---|---|
| Global Fatigue—NRS | Yes | Yes | • A single-item NRS of 0 to 10 points was used to measure the global degree of fatigue [54].<br>The recall period was determined during the past 7 days. |
| Pain intensity—NRS | Yes | Yes | • A single-item NRS of 0 to 10 points was used to measure the degree of pain intensity [55].<br>• The recall period was determined to be 24 h, both the worst and average paint intensity. |
| MAST | Yes | Yes | • The MAST consists of 8 items to measure medication adherence.<br>• This scale had good psychometric properties, with a Cronbach's α of 0.83 among chronic disease patients in Thailand [56]. |

Abbreviations: BRCS, Brief Resilient Coping Scale; DLQI, Dermatological Life Quality Index; EQ-5D-5L, EuroQoL-5 dimension-5 level; GAD-7, Generalized Anxiety Disorder 7-item; HRQoL, health-related quality of life; ISI, Insomnia Severity Index 7-item; MAST, Medication adherence using the Medication Adherence Scale in Thais; NA, not applicable; NRS, numerical rating scale; PHQ-9, Patient Health Questionnaire 9-item; PROs-CP, Patients-Reported Outcomes among People Living with Chronic Pruritus; PUSH-D, Patient Unique Stigmatization Holistic tool in Dermatology; RSES, Rosenberg Self-Esteem Scale; SSS-8, Somatic Symptom Scale 8-item; TAS-20, Toronto Alexithymia Scale 20-item; UCLA, University of California, Los Angeles; VAS, visual analogue scale; VRS, verbal rating scale; WHO, World Health Organization.

necessarily. Next, both translations will be combined by the two translators in conjunction with a team of experts who serve as independent observers. Any discrepancies that appear during this process will be resolved through team discussions.

Two independent bilingual native English speakers will independently blind-translate the synthesized measurement scales back into English. For the back-translation process, no translators had any medical health/dermatology background or acknowledged information regarding the particular original English version. To ensure that the translated version reflects the content of the original version, the content validity of the back-translation questionnaire will be reviewed and addressed. After that, both versions of the backward translations will be compared and reconciled with the original English version. To ensure equivalence between the original and translated Thai versions, conceptual equivalence (ensuring that words can hold the same conceptual meaning), item equivalence (individual items have the same relevance), and semantic equivalence (meaning of the items is the same in both version) will be employed. In the same manner as pilot testing and refinement of the scale development, a panel of experts and patients will review and refine the preliminary translated scales. Next, the EFA, non-parametric IRT, and initially psychometric analysis will be conducted to establish the validity, reliability, and scale stability (reproducibility) of the translated scales Thai version (i.e., Itchy-QoL, PUSH-D, and TAS-20).

To track the treatment responses, the responsiveness of the pruritus-specific disease activity control (planned to develop through this study) and pruritus-specific HRQoL (using the Itchy-QoL) in terms of the MCID will also be defined and established among patients with chronic pruritus. For scale interpretability in terms of MCID, we will incorporate both anchor-based and distributional criterion approaches using the change scales of disease activity and Itchy-QoL at week 12 of follow-up [31–33].

Based on the consecutive recruitment of the open cohort, we plan to establish a temporal validation by using the same population at different times [15]. After a six-month and one-year follow-up, we will reanalyze sets of PROs-CP measurement tools to reaffirm the psychometric properties of the particular scale for people living with chronic pruritus over time. Cross-validity and reliability will be analyzed in order to establish the scale's psychometric properties using an anchor-based questionnaire approach. Scale dimensionality for each

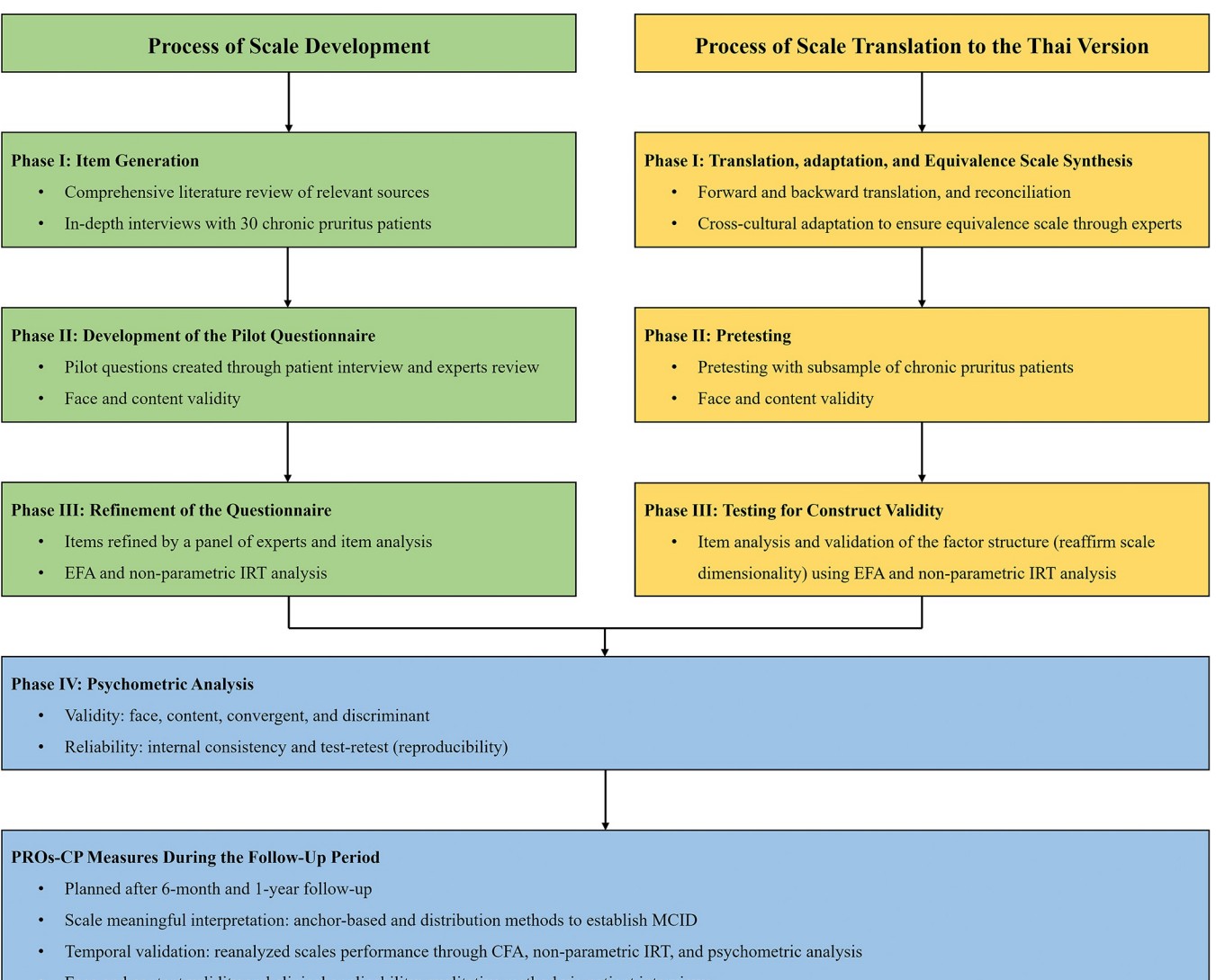

**Fig 2. Process of the scale development, translation, and psychometric validation of the PROs-CP measures.** Abbreviations: CFA, confirmatory factor analysis; EFA, exploratory factor analysis; IRT, item response theory; MCID, minimal clinically important difference; PROs-CP, Patients-Reported Outcomes among People Living with Chronic Pruritus.

multidimensional measurement tool will be reaffirmed using the confirmatory factor analysis (CFA) and non-parametric IRT methods. A quantitative method via face-to-face or video call using Zoom will also be employed. A random subsample (estimated approximately 50 patients) based on a one-on-one approach will be asked using structured and unstructured in-depth interviews to provide their perspectives on the face and content validity and relevancy of the PROs-CP tools. Moreover, based on feedback from patients and research personnel, the acceptability of scale measurement will be assessed through the study, along with the reasons for the unacceptability or unfavorable use of the PROs-CP measure in clinical practice.

## Study II: Patient perspectives on living with chronic pruritus

In this sub-studies, we will perform a qualitative study to explore the views and experiences of patients living with chronic pruritus. Regarding the purposive sampling, the subsample of

eligible patients will be invited through the participating open cohort recruiting. Ultimately, we plan to include patients based on maximum sample variation using key clinical characteristics (i.e., primary skin diseases, severity of symptoms, and burden or impact on daily life) to represent and reflect the perspective of people living with chronic pruritus. For convenience, either face-to-face or video calls using Zoom will be considered. A trained research assistant will conduct one-on-one qualitative interviews using structured and unstructured questions among Thai-speaking aged 20 years or older with chronic pruritus (approximately 60–120 min). Patients will be asked about the burden of chronic pruritus (e.g., which associated symptoms of pruritus they experienced and considered to be the worst), problems and concerns of how pruritus affected their lives, treatment benefits/responses, barriers/facilitators to achieving disease control, patient preferences, and how their perceived satisfaction with treatment care. As a part of quality control, lead investigators (SN and MC) will monitor the quality of the interview process and transcript and provide feedback to an interviewer as necessary.

## Study III: Longitudinal cohort and health outcomes

Based on an open registered cohort approach, eligible patients will be recruited continually to identify interested patients who intend to participate in the PROs-CP study. All patients will be monitored based on routine follow-up with a standard visit at outpatient clinics (approximately every 2–3 months). Data will be collected through the study follow-up periods every 2–3 months onwards, including pruritus characteristics and medical history (i.e., primary skin diseases, duration and localization of itch, IFSI classification and category of underlying skin disease, and symptoms associated with pruritus, comorbid conditions), physical examination, routine laboratory tests, sets of PROs measurement tools, treatment for specific-pruritus and non-pruritus, and healthcare utilization (i.e., direct medical costs, direct non-medical costs, and indirect costs). To facilitate longitudinal cohort retention and minimize missing data, various reminder strategies based on our routine follow-up care, case manager staff members will contact the eligible patients/caregivers through their preference contact approaches (i.e., phone, text message, or direct email) within 1–3 days before the visiting follow-up or anticipate data collection date.

All patients will be monitored for as long as possible. However, we plan to assess the severity and symptoms of pruritus, disease activity control, and longitudinal impact on health outcomes at one year of follow-up. These will include the impact of chronic pruritus and disease activity on long-term PRO outcomes (i.e., HRQoL, symptoms burden, and mental health and psychosocial issues), health utility, safety profiles (adverse events), hospitalization and healthcare visit, and healthcare expenditures (direct medical costs, direct non-medical costs, indirect costs, and out-of-pocket costs).

## Data management and security

Based on deidentified data, all data elements in this study cohort will be incorporated with the local joint data (electronic health records—outpatient and inpatient information and Pharmacy Dispensing and Laboratory Support System extract), as well as transcript data from the qualitative method will be entered continually into the PROs-CP database via the REDCap$^{TM}$ platform. However, the patient's identification information (i.e., name, telephone number, home address, email address, social media contacts, and family/caregiver member information) will not be entered into the database to protect the patient's privacy.

The external panel of two health information technicians from the Pharmacoepidemiology and Statistics Research Center will verify, cross-check, and monitor all data elements throughout the study follow-up periods to maintain high-quality information on the PROs-CP

database. Moreover, two data administrators will be checked for completeness and consistency with the source documents in a timely manner to minimize data errors and missing values. All data elements in the PROs-CP database will be managed and encrypted using a password code, which only the assigned management team can access to view and edit.

## Sample size estimation

Although we plan to include all eligible patients in this open longitudinal study, a pre-specified sample size for scale development and validation of the PRO-CP measurement tool was estimated. As such, sample size consideration for this study will be calculated based on two parameters. Firstly, we use a sample-to-item ratio (5–15 patients per question for both the EFA and CFA approach) to address a scale structure for factor analysis [8, 28, 29]. Secondly, a fair correlation coefficient statistic (>0.30) with anchor-based questions was used to establish convergent validity [34]. Regarding a power of 80% and 0.05 type I error, a minimum target of 396 patients with chronic pruritus is needed for this study cohort to account for missing responses of 20% during the cohort follow-up.

## Statistical analysis plan

All statistical analyses will be analyzed using Stata, version 16.0 (StataCorp, College Station, Texas). The significant level was tested at $P$ value <0.05 based on hypotheses of two-tailed tests. Missing values will be handled with a multiple imputation method. Descriptive statistics will be analyzed and presented as number (%), mean (SD), or median (min–max) as appropriate.

Item analysis will be assessed descriptively and summarized along with the normality of score distribution, including skewness and kurtosis tests. The Kaiser-Meyer-Olkin measure and Bartlett test of sphericity will be tested to ensure the appropriate scale factor analysis of the measurement scales. For preplanned EFA, we will employ a principle factor extraction method, with the obliquely rotated using the Promax criterion. Eigenvalues with a criterion of >1.0 will be considered as the number of factors retained. The unidimensional set or subscales will be tested using a non-parametric IRT. The item characteristics curve analysis will be visualized to describe the characteristics of individual items within the particular scale. During the follow-up period, we will perform a CFA with a maximum likelihood estimation to reaffirm the scale structure. Specific fit indices will be tested, including the comparative-fit index (>0.900), non-normed fit index or Tucker-Lewis index (<0.900), standardized root mean squared residual (<0.100), and root mean square error of approximation (<0.100).

To assess the psychometric properties, we will test the convergent validity of the sets of PROs-CP measurement scales using Pearson's or Spearman's correlation coefficients as appropriate. The 95% confidence intervals (CIs) of correlation coefficient statistics will be estimated using the bootstrap resampling method. A know-group or discriminant validity will be assessed based on pruritus intensity—visual analog scale as follows: mild (0.1–2.9), moderate (3.0–6.9), severe (7.0–8.9), and very severe (9.0–10.0) [2]. Internal consistency reliability will be tested using Cronbach's α and McDonald's ω coefficients with 95% CIs (values of >0.7 indicate acceptable reliability). Scale reproducibility will be tested using intraclass correlation coefficients. The kappa (κ) coefficient of the agreement and area under the receiver operating characteristic curve will be used to assess the optimal cut-off severity or burden score to establish banding of measurement scales with anchor-based questions.

All interviews' audio recordings will be transcribed verbatim. Deidentified transcript data from the in-depth interview process will be uploaded, coded, and refined by professional experts in qualitative research (RA, CP, and YR). For qualitative analysis, we will employ

content analysis to summarize the interview information and assess the consistency of coding and refinement concepts that capture patient views and experiences of chronic pruritus. A thematic approach based on grounded theory techniques will be used to analyze and identify patterns or themes (subthemes) from patient perspective information [35, 36]. Furthermore, we also plan to qualitatively synthesize among subgroups with severe or very severe pruritus intensity to reflect the impacts of chronic pruritus that are most important to patients with high-burden disease activity. The saturation of concept findings from qualitative interviews will be reaffirmed using saturation tables to verify and track the concepts of how pruritus affected the patient's life.

For longitudinal follow-up at one year, differences between groups (predefined based on pruritus intensity as mild, moderate, severe, and very severe) will be tested using Fisher's exact test for categorical data and analysis of covariance or Kruskal-Wallis test for continuous data. The multivariable Tobit, modified Poisson, and ordinal logistic regression analysis will be analyzed to explain the relationship between the degree of pruritus intensity and long-term health outcomes. The effect estimates of the findings will be reported as β coefficients, risk ratios, and common odds ratios, respectively, corresponding with CIs.

## Ethical consideration and dissemination of the findings

**Ethical consideration.** The study protocol (version 3, March 6, 2024) was approved by the Ethics Committee of the Faculty of Medicine, Chiang Mai University (MED-2566-0299). The PROs-CP cohort will be conducted according to the Declaration of Helsinki as well as the amendments or comparable ethical standards. Written informed consent will be obtained from all patients before participating in the study. Patients will not receive financial or non-financial compensation.

All patients may withdraw their informed consent at any time during the study period. In the case of informed consent withdrawal, patients will have no effect on routine clinical management at the study site. All data collected is securely kept separate in a locked filing cabinet and will remain confidential and shared only with the authorized research team.

**Dissemination of the findings.** Any protocol amendments or justifications will be reported in the final reports for study transparency. This study protocol was followed by adapting the international, consensus-based, PRO-specific protocol guidance (the SPIRIT-PRO Extension) [37]. To disseminate the findings, we will report the study results to follow the Reporting of studies Conducted using Observational Routinely-collected health Data (RECORD) statement [38]. The reporting of the results with the psychometric properties analyses will be supplemented by the COnsensus-based Standards for the selection of health status Measurement INstruments (COSMIN) reporting guideline [39]. Moreover, the findings regarding the qualitative approach will be reported in line with the Standards for Reporting Qualitative Research (SRQR) statement [40]. We plan to disseminate the findings from this study via scientific communities and publications in peer-reviewed scientific journals.

## Discussion

Although there has been an improvement in dermatology research and the development of promising novel pharmacological therapy in recent years [41–43], the burden of chronic pruritus remains under-recognized in clinical practice. Indeed, partly or narrow set aspects of disease assessment may limit the overarching patient experiences, especially in patients with a high burden of pruritus disease and symptoms activity. Moreover, to date, little is known about the comprehensive perspectives of PRO measures on long-term outcomes among people living with chronic pruritus.

To the best of our knowledge, this is the first comprehensive set of PRO measures among patients with chronic pruritus based on a prospective longitudinal study. This study will draw evidence-based regarding the strengths of both quantitative and qualitative approaches using multistage, mixed-methods parallel designs. This study will provide detailed and nuanced information regarding the clinical utility of adopted PROs in daily dermatology practice. Taken together, we postulated that long-term PRO monitoring may help to better understand the burden of pruritus and help to optimize treatment targeted with respect to patient perspectives. However, owing to a single-center, university hospital-based cohort, this study may be limited and generalized to other settings with different healthcare systems or chronic skin disease populations should be caution.

## Cohort status

Currently, the validated non-Thai version tools (ItchyQoL, PUSH-D, and TAS-20) have been translated. Meanwhile, the measurement tools for chronic pruritus-specific disease activity control and treatment satisfaction are now under phase I—item generation of scale development. As of October 30, 2024, the PROs-CP cohort continually enrolls patients by invitation. A total of 70 eligible patients have participated in this cohort.

## Conclusions

This open cohort, prospective longitudinal study will track PROs and reflect the views and experiences of Thai people living with chronic pruritus using a mixed-methods approach. To supplement clinical outcome assessment in daily practice and help guide effective interventions, findings from the PROs-CP cohort will further leverage evidence regarding comprehensive sets of PRO measures based on the perspectives of patients with chronic pruritus. These findings also inform the utility and applicability of PRO measures and long-term outcomes among patients with chronic pruritus. Our findings will be disseminated through scientific conferences and publications in peer-reviewed journals.

## Acknowledgments

We thank all study patients, research personnel, and staff involved in the study cohort. Particular thanks to Arun Kunti, research assistant at Pharmacoepidemiology and Statistics Research Center (PESRC), Faculty of Pharmacy, Chiang Mai University, for contributing to the project management.

## Author Contributions

**Conceptualization:** Surapon Nochaiwong, Mati Chuamanochan.

**Funding acquisition:** Surapon Nochaiwong, Mati Chuamanochan.

**Methodology:** Surapon Nochaiwong, Chidchanok Ruengorn, Salin Kiratikanon, Rujira Ruji-wetpongstorn, Panjit Chieosilapatham, Napatra Tovanabutra, Siri Chiewchanvit, Ratana-porn Awiphan, Chabaphai Phosuya, Yongyuth Ruanta, Kednapa Thavorn, Mati Chuamanochan.

**Supervision:** Surapon Nochaiwong, Mati Chuamanochan.

**Writing – original draft:** Surapon Nochaiwong, Mati Chuamanochan.

**Writing – review & editing:** Chidchanok Ruengorn, Salin Kiratikanon, Rujira Rujiwetpong-storn, Panjit Chieosilapatham, Napatra Tovanabutra, Siri Chiewchanvit, Ratanaporn Awi-phan, Chabaphai Phosuya, Yongyuth Ruanta, Kednapa Thavorn.

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
