## [Decision Letter · Decision Letter 0]

28 Oct 2024

PONE-D-24-14361Patient-Reported Outcomes Among People Living with Chronic Pruritus (PROs-CP): Protocol for a Single-Center, Multistage, Mixed-Methods Prospective Cohort Study in ThailandPLOS ONE

 Dear Dr. Chuamanochan,

Thank you for submitting your manuscript to PLOS ONE. After careful consideration, we feel that it has merit but does not fully meet PLOS ONE’s publication criteria as it currently stands. Therefore, we invite you to submit a revised version of the manuscript that addresses the points raised during the review process.

**ACADEMIC EDITOR: After a careful reading of the reviewers comments I recommend to the authors to revise the manuscript based on the PLOS ONEs publication criteria.****The changes recommended by reviewers are important to improve the scientific merit of the manuscript****best regards****José Luiz Vieira**==============================

We look forward to receiving your revised manuscript.

Kind regards,

José Luiz Fernandes Vieira

Academic Editor

PLOS ONE

“This work was supported by Pharmacoepidemiology and Statistics Research Center (PESRC) and partially supported by Faculty of Medicine Research Fund (grant No. MED 74/2567), Chiang Mai University, Thailand.”

3. In the online submission form, you indicated that [Data from the study will be made available at the end of the study, on request. Requests will be subject to approval by the PROs-CP chief investigator, the advisory committee, and the relevant ethical bodies.].

Additional Editor Comments:

On Page 6:

“…..To the best of our knowledge, although there have been well-validated PRO measurement tools in dermatology research?, there is limited evidence of the adopted comprehensive aspects of PRO measures in long-term follow-up among people living with chronic pruritus….

? Can you give reference(s)

2. Page 10:

…..Of these, one person will have knowledge of dermatology and be aware of the study’s purpose and objectives, while the other layperson will not necessary…..

Can you clear if the person will be a dermatologist and one of the authors? If not, explain what you mean by a person with knowledge of dermatology?

3. Rephrase the line:

During this process, neither translator will be aware of any background information regarding the original English version, the study’s purpose, or any medical background.

4. Duration of the study can be specified in Methods section also.

5. Clarify the structure of the interviews and how consistency will be maintained.

6. Have the authors devise any strategy to minimize loss to follow-up, explain how missing data will be handled in the analysis

7. Can we Simplify the description of statistical tests and focus on those most critical to the study with a brief justification for each chosen method.

Reviewers' comments:

Reviewer's Responses to Questions

**Comments to the Author**

1. Does the manuscript provide a valid rationale for the proposed study, with clearly identified and justified research questions?

Reviewer #1: Yes

Reviewer #2: Yes

2. Is the protocol technically sound and planned in a manner that will lead to a meaningful outcome and allow testing the stated hypotheses?

Reviewer #1: Yes

Reviewer #2: Yes

3. Is the methodology feasible and described in sufficient detail to allow the work to be replicable?

Reviewer #1: Yes

Reviewer #2: Yes

4. Have the authors described where all data underlying the findings will be made available when the study is complete?

Reviewer #1: No

Reviewer #2: Yes

5. Is the manuscript presented in an intelligible fashion and written in standard English?

Reviewer #1: Yes

Reviewer #2: Yes

6. Review Comments to the Author

You may also provide optional suggestions and comments to authors that they might find helpful in planning their study.

Reviewer #1: The authors are appreciated for acknowledging the need to develop new instruments for measuring disease activity control and treatment satisfaction for CP.

Although being a single-centre study may limit the generalisability of the findings but still it is a well curated study. Following are few suggestions:

1. On Page 6:

“…..To the best of our knowledge, although there have been well-validated PRO measurement tools in dermatology research?, there is limited evidence of the adopted comprehensive aspects of PRO measures in long-term follow-up among people living with chronic pruritus….

? Can you give reference(s)

2. Page 10:

…..Of these, one person will have knowledge of dermatology and be aware of the study’s purpose and objectives, while the other layperson will not necessary…..

Can you clear if the person will be a dermatologist and one of the authors? If not, explain what you mean by a person with knowledge of dermatology?

3. Rephrase the line:

During this process, neither translator will be aware of any background information regarding the original English version, the study’s purpose, or any medical background.

4. Duration of the study can be specified in Methods section also.

5. Clarify the structure of the interviews and how consistency will be maintained.

6. Have the authors devise any strategy to minimize loss to follow-up, explain how missing data will be handled in the analysis

7. Can we Simplify the description of statistical tests and focus on those most critical to the study with a brief justification for each chosen method.

The proposed study is comprehensive and well-structured, addresses an important gap in dermatological research.

Overall, the suggestion would be to the refine presentation of the methodology and focus on clarity and conciseness. Try to be succinct and less verbose.

Reviewer #2: This is an innovative study design combining a longitudinal study and a mixed methods design for developing a standard set of PRO measurement tools for CP. Your study protocol seems very well structured following a profound literature review.

7. PLOS authors have the option to publish the peer review history of their article (what does this mean?). If published, this will include your full peer review and any attached files.

Reviewer #1: No

Reviewer #2: **Yes: **Sabine Steinke

---

## [Author Response · Author response to Decision Letter 0]

31 Oct 2024

Response to Reviewers

PONE-D-24-14361—entitled, “Patient-Reported Outcomes Among People Living with Chronic Pruritus (PROs-CP): Protocol for a Single-Center, Multistage, Mixed-Methods Prospective Cohort Study in Thailand”

Journal Requirements

#1. Please ensure that your manuscript meets PLOS ONE’s style requirements, including those for file naming.

We have carefully checked throughout the manuscript.

#2. Please state what role the funders took in the study. 

Thank you for your suggestion. We have stated the role funders as follows:

“This work was supported by the Pharmacoepidemiology and Statistics Research Center (PESRC) and partially supported by the Faculty of Medicine Research Fund (grant No. MED 74/2567), Chiang Mai University, Thailand. The funders had no role in study design, data collection and analysis, decision to publish, or preparation of the manuscript.”

#3. In the online submission form, you indicated that [Data from the study will be made available at the end of the study, on request. Requests will be subject to approval by the PROs-CP chief investigator, the advisory committee, and the relevant ethical bodies.].

Thank you for your concerns. This manuscript was prepared based on the study protocol, and no data was generated during this work. To make it more clear, we have revised and stated this issue as follows: 

“No datasets were generated or analyzed during the current study, so the data availability policy is not applicable. However, all relevant data from this study will be made available under open access upon study completion.”

#4. Please review your reference list to ensure that it is complete and correct. If you have cited papers that have been retracted, please include the rationale for doing so in the manuscript text, or remove these references and replace them with relevant current references. Any changes to the reference list should be mentioned in the rebuttal letter that accompanies your revised manuscript. If you need to cite a retracted article, indicate the article’s retracted status in the References list and also include a citation and full reference for the retraction notice.

We have carefully cross-checked the completeness and correctness of the references cited, and no retracted articles have been included in this manuscript.

Additional Editor Comments

#1. “…..To the best of our knowledge, although there have been well-validated PRO measurement tools in dermatology research?, there is limited evidence of the adopted comprehensive aspects of PRO measures in long-term follow-up among people living with chronic pruritus….

? Can you give reference(s)

References have been added as recommended.

#2. Page 10:

…..Of these, one person will have knowledge of dermatology and be aware of the study’s purpose and objectives, while the other layperson will not necessary…..

Can you clear if the person will be a dermatologist and one of the authors? If not, explain what you mean by a person with knowledge of dermatology?

Thank you for your concerns. For clarification, we have rewritten this sentence as follows: 

“Of these, one person will be a board-certified dermatologist (MC), who is the study investigator and recognizes the study’s purpose and objectives, while the other layperson will not necessarily.”

#3. Rephrase the line:

During this process, neither translator will be aware of any background information regarding the original English version, the study’s purpose, or any medical background.

Thank you for your suggestion. We have rewritten this sentence as recommended. 

“For the back-translation process, no translators had any medical health/dermatology background or acknowledged information regarding the particular original English version.”

#4. Duration of the study can be specified in Methods section also.

Thank you for your pertinent observation. Regarding an open cohort design approach, ultimately, eligible patients will be continually to follow-up as long as possible (at least 1-year follow-up). We have addressed this issue under the “Materials and Methods—Study III: Longitudinal Cohort and Health Outcomes” section, as follows:

“Based on an open registered cohort approach, eligible patients will be recruited continually to identify interested patients who intend to participate in the PROs-CP study. All patients will be monitored based on routine follow-up with a standard visit at outpatient clinics (approximately every 2–3 months).”

AND

“All patients will be monitored for as long as possible. However, we plan to assess the severity and symptoms of pruritus, disease activity control, and longitudinal impact on health outcomes at one year of follow-up.”

#5. Clarify the structure of the interviews and how consistency will be maintained.

Thank you very much for your concerns. Regarding the qualitative approach, one trained research assistant will conduct the interview processes through one-on-one interviews using structured and unstructured questions. We have addressed this issue under the “Materials and Methods—Study II: Patient Perspectives on Living with Chronic Pruritus” section, as follows:

“A trained research assistant will conduct one-on-one qualitative interviews using structured and unstructured questions among Thai-speaking aged 20 years or older with chronic pruritus (approximately 60-120 min). Patients will be asked about the burden of chronic pruritus (e.g., which associated symptoms of pruritus they experienced and considered to be the worst), problems and concerns of how pruritus affected their lives, treatment benefits/responses, barriers/facilitators to achieving disease control, patient preferences, and how their perceived satisfaction with treatment care. As a part of quality control, lead investigators (SN and MC) will monitor the quality of the interview process and transcript and provide feedback to an interviewer as necessary.”

#6. Have the authors devise any strategy to minimize loss to follow-up, explain how missing data will be handled in the analysis.

Thank you very much for your insightful comments. According to the observation prospective cohort design, no specific interventions/strategies were employed in this study. Based on our routine follow-up care, the case manager staff member will communicate with patients/caregivers regarding the study’s purposes/objectives to minimize patient dropout and missing data.

Regarding your concerns, we have added this issue under the Study III: Longitudinal Cohort and Health Outcomes and Statistical Analysis Plan sections, as follows:

Under the Study III: Longitudinal Cohort and Health Outcomes

“To facilitate longitudinal cohort retention and minimize missing data, various reminder strategies based on our routine follow-up care, case manager staff members will contact the eligible patients/caregivers through their preference contact approaches (i.e., phone, text message, or direct email) within 1-3 days before the visiting follow-up or anticipate data collection date.”

AND

Under the Statistical Analysis Plan

“Missing values will be handled with a multiple imputation method.”

#7. Can we Simplify the description of statistical tests and focus on those most critical to the study with a brief justification for each chosen method

Thank you very much for your advice on simplifying and adjusting the statistical part to be more succinct. We plan to conduct quantitative and qualitative research using a mixed-methods approach. Subsequently, data analyses will comprise both quantitative and qualitative approaches separately. Moreover, we acknowledge the importance of quality for statistical methods reporting in line with the SPIRIT-PRO Extension, RECORD statement, COSMIN guideline, and SRQR reporting statement. In this circumstance, we try to describe and believe that the “Statistical Analysis Plan” was reported based on a minimum set to address the high-quality methods reporting, which accounts for the completeness and conciseness aspects. 

References

- Calvert M, et al. Guidelines for Inclusion of Patient-Reported Outcomes in Clinical Trial Protocols: The SPIRIT-PRO Extension. JAMA. 2018;319(5):483-94.

- Benchimol EI, et al. The REporting of studies Conducted using Observational Routinely-collected health Data (RECORD) statement. PLoS Med. 2015;12(10):e1001885.

- Gagnier JJ, Lai J, Mokkink LB, Terwee CB. COSMIN reporting guideline for studies on measurement properties of patient-reported outcome measures. Qual Life Res. 2021;30(8):2197-218.

- O'Brien BC, et al. Standards for reporting qualitative research: a synthesis of recommendations. Acad Med. 2014;89(9):1245-51.

Review Comments to the Author

Reviewer 1

The authors are appreciated for acknowledging the need to develop new instruments for measuring disease activity control and treatment satisfaction for CP. Although being a single-centre study may limit the generalisability of the findings but still it is a well curated study. Following are few suggestions:

#1. On Page 6:

“…..To the best of our knowledge, although there have been well-validated PRO measurement tools in dermatology research?, there is limited evidence of the adopted comprehensive aspects of PRO measures in long-term follow-up among people living with chronic pruritus….

? Can you give reference(s)

References have been added as recommended.

#2. Page 10:

…..Of these, one person will have knowledge of dermatology and be aware of the study’s purpose and objectives, while the other layperson will not necessary…..

Can you clear if the person will be a dermatologist and one of the authors? If not, explain what you mean by a person with knowledge of dermatology?

Thank you for your concerns. For clarification, we have rewritten this sentence as follows: 

“Of these, one person will be a board-certified dermatologist (MC), who is the study investigator and recognizes the study’s purpose and objectives, while the other layperson will not necessarily.”

3. Rephrase the line:

During this process, neither translator will be aware of any background information regarding the original English version, the study’s purpose, or any medical background.

Thank you for your suggestion. We have rewritten this sentence as recommended. 

“For the back-translation process, no translators had any medical health/dermatology background or acknowledged information regarding the particular original English version.”

4. Duration of the study can be specified in Methods section also.

Thank you for your pertinent observation. Regarding an open cohort design approach, ultimately, eligible patients will be continually to follow-up as long as possible (at least 1-year follow-up). We have addressed this issue under the “Materials and Methods—Study III: Longitudinal Cohort and Health Outcomes” section, as follows:

“Based on an open registered cohort approach, eligible patients will be recruited continually to identify interested patients who intend to participate in the PROs-CP study. All patients will be monitored based on routine follow-up with a standard visit at outpatient clinics (approximately every 2–3 months).”

AND

“All patients will be monitored for as long as possible. However, we plan to assess the severity and symptoms of pruritus, disease activity control, and longitudinal impact on health outcomes at one year of follow-up.”

5. Clarify the structure of the interviews and how consistency will be maintained.

Thank you very much for your concerns. Regarding the qualitative approach, one trained research assistant will conduct the interview processes through one-on-one interviews using structured and unstructured questions. We have addressed this issue under the “Materials and Methods—Study II: Patient Perspectives on Living with Chronic Pruritus” section, as follows:

“A trained research assistant will conduct one-on-one qualitative interviews using structured and unstructured questions among Thai-speaking aged 20 years or older with chronic pruritus (approximately 60-120 min). Patients will be asked about the burden of chronic pruritus (e.g., which associated symptoms of pruritus they experienced and considered to be the worst), problems and concerns of how pruritus affected their lives, treatment benefits/responses, barriers/facilitators to achieving disease control, patient preferences, and how their perceived satisfaction with treatment care. As a part of quality control, lead investigators (SN and MC) will monitor the quality of the interview process and transcript and provide feedback to an interviewer as necessary.”

6. Have the authors devise any strategy to minimize loss to follow-up, explain how missing data will be handled in the analysis

Thank you very much for your insightful comments. According to the observation prospective cohort design, no specific interventions/strategies were employed in this study. Based on our routine follow-up care, the case manager staff member will communicate with patients/caregivers regarding the study’s purposes/objectives to minimize patient dropout and missing data.

Regarding your concerns, we have added this issue under the Study III: Longitudinal Cohort and Health Outcomes and Statistical Analysis Plan sections, as follows:

Under the Study III: Longitudinal Cohort and Health Outcomes

“To facilitate longitudinal cohort retention and minimize missing data, various reminder strategies based on our routine follow-up care, case manager staff members will contact the eligible patients/caregivers through their preference contact approaches (i.e., phone, text message, or direct email) within 1-3 days before the visiting follow-up or anticipate data collection date.”

AND

Under the Statistical Analysis Plan

“Missing values will be handled with a multiple imputation method.”

7. Can we Simplify the description of statistical tests and focus on those most critical to the study with a brief justification for each chosen method.

The proposed study is comprehensive and well-structured, addresses an important gap in dermatological research.

Overall, the suggestion would be to the refine presentation of the methodology and focus on clarity and conciseness. Try to be succinct and less verbose.

Thank you very much for your advice on simplifying and adjusting the statistical part to be more succinct. We plan to conduct quantitative and qualitative research using a mixed-methods approach. Subsequently, data analyses will comprise both quantitative and qualitative approaches separately. Moreover, we acknowledge the importance of quality for statistical methods reporting in line with the SPIRIT-PRO Extension, RECORD statement, COSMIN guideline, and SRQR reporting statement. In this circumstance, we try to describe and believe that the “Statistical Analysis Plan” was reported based on a minimum set to address the high-quality methods reporting, which accounts for the completeness and conciseness aspects. 

References

- Calvert M, et al. Guidelines for Inclusion of Patient-Reported Outcomes in Clinical Trial Protocols: The SPIRIT-PRO Extension. JAMA. 2018;319(5):483-94.

- Benchimol EI, et al. The REporting of studies Conducted using Observational Routinely-collected health Data (RECORD) statement. PLoS Med. 2015;12(10):e1001885.

- Gagnier JJ, Lai J, Mokkink LB, Terwee CB. COSMIN reporting guideline for studies on measurement properties of patient-reported outcome measures. Qual Life Res. 2021;30(8):2197-218.

- O'Brien BC, et al. Standards for reporting qualitative research: a synthesis of recommendations. Acad Med. 2014;89(9):1245-51.

Revi

---

## [Editor Report · Decision Letter 1]

13 Nov 2024

Patient-Reported Outcomes Among People Living with Chronic Pruritus (PROs-CP): Protocol for a Single-Center, Multistage, Mixed-Methods Prospective Cohort Study in Thailand

PONE-D-24-14361R1

Dear Dr. Mati Chuamanochan

We’re pleased to inform you that your manuscript has been judged scientifically suitable for publication and will be formally accepted for publication once it meets all outstanding technical requirements.

Kind regards,

José Luiz Fernandes Vieira

Academic Editor

PLOS ONE

Additional Editor Comments (optional): All the reviewers suggestion were done by the authors.

Congratulations

José Luiz Vieira
---

## [Editor Report · Acceptance letter]

15 Nov 2024

PONE-D-24-14361R1 

PLOS ONE

Dear Dr. Chuamanochan, 

I'm pleased to inform you that your manuscript has been deemed suitable for publication in PLOS ONE. Congratulations! Your manuscript is now being handed over to our production team.

Kind regards, 

on behalf of

Dr. José Luiz Fernandes Vieira 

Academic Editor

PLOS ONE